

# Influence of Various Criteria on Identifying the Springtime Tropospheric Ozone Depletion Events (ODEs) at Utqiaġvik, Arctic

Xiaochun Zhu[1], Le Cao[1], Xin Yang[2], Simeng Li[3], Jiandong Wang[1], and Tianliang Zhao[1]

[1]China Meteorological Administration Aerosol-Cloud and Precipitation Key Laboratory, Nanjing University of Information Science and Technology, Nanjing, 210044, China
[2]British Antarctic Survey, Natural Environment Research Council, Cambridge, UK
[3]Institute of Environmental Sciences, Universiteit Leiden, Leiden, 2333 CA, the Netherlands

**Correspondence:** L. Cao
(le.cao@nuist.edu.cn)

**Abstract.** Tropospheric ozone depletion events (ODEs) occurring in the Arctic spring are a unique photochemical phenomenon in which the boundary layer ozone drops rapidly to near-zero levels. However, the criterion for identifying ODEs remains inconsistent among different studies, which may influence conclusions regarding the characteristics of ODEs. To address this issue, in this study, we applied various criteria used in previous studies to identify springtime ODEs at Utqiaġvik, Arctic (the BRW station), based on observational data spanning 23 years (2000-2022), and investigated the influences of implementing different criteria. We compared three types of criteria: traditional methods (fixed thresholds), variability-based methods (considering mean and standard deviation), and machine learning methods (Isolation Forest), and found that criteria using fixed thresholds (e.g., 10 ppbv) and relative thresholds based on monthly average ozone levels (0.42 times the monthly average) are more suitable for capturing ODEs at BRW compared to other criteria. Results applying these appropriate criteria all reveal a significant decline in ODE occurrence frequency over the investigated 23 years, particularly in April, suggesting potential links to climate change and Arctic sea ice melting. However, implementing relative thresholds or more stringent thresholds (5 ppbv and 4 ppbv) instead of the 10 ppbv threshold would display a more significant decline in the number of ODE hours across these 23 years. Further investigation of meteorological conditions indicates that ODEs at BRW are more prevalent under northerly and northeasterly winds with moderate wind speeds (3-6 m s$^{-1}$), at lower temperatures, and higher pressures, while severe ODEs are more associated with lower wind speeds and temperatures below 256 K. This research highlights the importance of selecting appropriate criteria to accurately identify ODEs and contributes to a better understanding of the complex processes driving the Arctic ODEs.

## 1 Introduction

The Arctic has been described as an important "window" on the global environment, as changes in the Arctic serve as a precursor for the global changes anticipated as the Earth's temperature rises (Thoman et al., 2023). Among all the changes in the Arctic environment, the variation in the ozone concentration during the spring seasons has attracted considerable attention from the scientific community. In the stratosphere, the depletion of ozone in the Arctic spring, which is caused due to the



existence of polar stratospheric clouds and halogen compounds (Crutzen and Arnold, 1986; Cairo and Colavitto, 2020; Seinfeld and Pandis, 2006; Akimoto, 2016), allows for increased UV radiation that can lead to higher rates of skin cancer, cataracts,

and weakened immune systems in humans (Umar and Tasduq, 2022), while also affecting marine and terrestrial ecosystems, potentially altering weather patterns and contributing to climate change (Liu et al., 2022).

In contrast to the stratospheric ozone depletion, in the lower troposphere, a unique phenomenon, namely ozone depletion events (ODEs), has also been frequently observed in the near-surface layer of the Arctic since the 1980s (Oltmans, 1981; Barrie et al., 1988; Bottenheim et al., 1986; Anlauf et al., 1995; Bottenheim et al., 2002; Shupe et al., 2022). This phenomenon is

associated with a photochemical process that can activate halogen ions (i.e., $Br^-$, $Cl^-$, $I^-$) from substrates such as ice/snow packs (Lehrer et al., 2004; Simpson et al., 2007b; Abbatt et al., 2012; Pratt et al., 2013; Custard et al., 2017) and sea-salt aerosols (Michalowski et al., 2000; Yang et al., 2010, 2019, 2020; Thomas et al., 2011, 2012; Huang et al., 2020) into reactive halogens that can deplete ozone. Consequently, the ozone in the polar boundary layer frequently falls from background levels (30-40 ppbv) to less than 10 ppbv or even near-zero values within a few days or even hours in the springtime of the Arctic.

The occurrence of ODEs is related to a complex photochemical process as follows (Simpson et al., 2007b),

$$X_2 + h\nu \rightarrow 2X,$$
$$2X + 2O_3 \rightarrow 2XO + 2O_2,$$
$$\underline{XO + XO \rightarrow X_2(2X) + O_2,}$$
$$\text{Net}: 2O_3 + h\nu \rightarrow 3O_2. \tag{R1}$$

In reaction cycle (R1), X denotes halogen species (i.e., Br, Cl and I). When the sun rises in the Arctic spring, halogen containing compounds (i.e., $X_2$) in the atmosphere are photodissociated into halogen atoms (i.e., X). These halogen atoms subsequently react rapidly with ozone, producing halogen monoxide XO, as depicted in reaction (R1). XO then participates in self-reactions

that regenerate X atoms, thereby depleting ozone without consuming halogens.

However, in reaction cycle (R1), the total amount of halogens remains constant, which is inconsistent with measurements at Arctic coastal stations in which a substantial increase in halogen concentrations is observed during ODEs (Hausmann and Platt, 1994; Zhao et al., 2016). Furthermore, previous researches have demonstrated that ozone cannot be depleted in such a short time if only gas-phase reactions occur (Lehrer et al., 2004). Thus, another reaction cycle including heterogeneous reactions

was proposed to explain the rapid ozone decline and the large amount of reactive halogens in the atmosphere during ODEs (Fan and Jacob, 1992; McConnell et al., 1992; Platt and Lehrer, 1997; Tang and McConnell, 1996; Wennberg, 1999),. Using bromine as a representation of halogen species, this proposed reaction cycle can be outlined as follows:

$$BrO + HO_2 \rightarrow HOBr + O_2,$$
$$HOBr + H^+ + Br^- \xrightarrow{mp} Br_2 + H_2O,$$
$$Br_2 + h\nu \rightarrow 2Br,$$
$$\underline{Br + O_3 \rightarrow BrO + O_2,}$$
$$\text{Net}: O_3 + HO_2 + H^+ + Br^- + h\nu \xrightarrow{mp} 2O_2 + Br + H_2O. \tag{R2}$$



In reaction cycle (R2), hypobromous acid (HOBr) can activate bromides (Br$^-$) from substrates such as ice, snowpack, and
sea-salt aerosols, leading to the conversion of bromides into reactive halogen species such as Br$_2$, which in turn undergo
photodissociation and consume ozone. Consequently, ozone is continuously depleted and the total bromine amount in the
atmosphere is explosively elevated during ODEs. Thus, this process is referred to as the "bromine explosion mechanism" (Platt
and Lehrer, 1997; Wennberg, 1999).

ODEs can exert profound influence on both the Arctic environment and the global ecosystems. First, they can alter the
radiation balance in the Arctic by reducing the infrared radiations absorbed by the atmosphere (Lacis et al., 1990). Moreover,
during ODEs, the oxidative capacity of the Arctic atmosphere is dominated by the enhanced bromine compounds. This shift
in the oxidative capacity facilitates the oxidation of elemental mercury (Hg(0)), followed by deposition of active mercury
(Hg(II)), a pollutant that is highly toxic to humans. The increased deposition of active mercury would ultimately jeopardize
the human health in mid-latitude regions through snow melting and oceanic circulations (Ariya et al., 2004; Pratt et al., 2013;
Dastoor et al., 2022; Basu et al., 2022).

Since the discovery of ODEs, our understanding of this phenomenon has been improved, such as the role of sea ice formation
(Jones et al., 2011; Lehrer et al., 2004; Simpson et al., 2007a; Abbatt et al., 2012; Peterson et al., 2019), the linkage to climate
variability (Koo et al., 2014), the influence of snowpack photochemical emissions (Pratt et al., 2013; Custard et al., 2017;
Herrmann et al., 2021, 2022) and the Arctic blowing snow (Chen et al., 2022; Yang et al., 2010, 2019, 2020; Huang et al.,
2020), as well as the connection to the total column ozone (Cao et al., 2022). However, to date, the criterion for identifying
ODEs remains unclear. In previous studies, the occurrence of ODEs was mostly recognized by the volume mixing ratio of
the surface ozone, when the surface ozone decreases to a level below a fixed threshold. However, the threshold is inconsistent
across different studies, ranging from 20 ppbv down to 4 ppbv. For example, Tarasick and Bottenheim (2002) and Koo et al.
(2012) used a threshold of 10 ppbv, while Bottenheim et al. (2009), Frieß et al. (2011) and Jacobi et al. (2010) utilized a 5 ppbv.
Meanwhile, Piot and Von Glasow (2008) and Ridley et al. (2003) identified ODEs using a more stringent 4 ppbv threshold.
In contrast to these studies employing a fixed threshold, Cao et al. (2022) used a criterion depending on the mean value and
the standard deviation of the surface ozone across various months and years to identify ODEs. This criterion was based on the
method of Bian et al. (2018) which was originally used to indicate uncommon variations in the surface ozone in polar regions.
In that study, Cao et al. (2022) applied this criterion to pick out ODE hours from surface ozone measurements at Halley Station
in Antarctica for the spring months from 2007 to 2013, and the results demonstrated that the criterion is effective in identifying
ODEs from the springtime surface ozone measurements at Halley Station.

Because the identification criterion for ODEs may influence conclusions regarding the characteristics of ODEs, such as the
interannual variability of ODEs and the relationship between meteorological conditions and the occurrence frequency of ODEs,
in this study, we employed various criteria to identify ODEs from 23 years of spring observational data detected at Utqiaġvik
in Alaska, and explored the impacts of using different screening criteria on the results. We then selected appropriate criteria
to investigate the correlation between meteorological parameters (e.g., wind speed, wind direction, temperature, and pressure)
and the occurrence of ODEs, and compared the results to assess the impact of different criteria on the conclusions.



**Table 1.** Criteria used to identify ODEs in this study and their expressions.

| Type | Name | Formula |
|------|------|---------|
| Traditional Methods | TM1 | $([O_3]_i < 10\,\mathrm{ppbv}) \cap ([O_3]_{i+1} < 10\,\mathrm{ppbv})$ |
| | TM1-5 ppbv | $([O_3]_i < 5\,\mathrm{ppbv}) \cap ([O_3]_{i+1} < 5\,\mathrm{ppbv})$ |
| | TM1-4 ppbv | $([O_3]_i < 4\,\mathrm{ppbv}) \cap ([O_3]_{i+1} < 4\,\mathrm{ppbv})$ |
| | TM2 | $([O_3]_i < 10\,\mathrm{ppbv}) \cap ([O_3]_{i+j} < 10\,\mathrm{ppbv}), j \in \{1,2,3,4,5,6\}$ |
| | TM3 | $([O_3]_i < \beta\overline{[O_3]}) \cap ([O_3]_{i+1} < \beta\overline{[O_3]}), \beta = 0.42$ |
| | TM4 | $([O_3]_i < \beta\overline{[O_3]}) \cap ([O_3]_{i+j} < \beta\overline{[O_3]}), \beta = 0.42, j \in \{1,2,3,4,5,6\}$ |
| | TM5 | $\mathrm{TM4} \cup (-\frac{d[O_3]}{dt} > 1\,\mathrm{ppbv\ h^{-1}})$ |
| Variability-Based Method | VM | $[O_3]_i - \overline{[O_3]} < \alpha \cdot \sigma, \alpha = -1.5$ |
| Machine Learning Method | Isolation Forest (IF) | - |

## 2 Observational Data and Screening Criteria

We first used different criteria to screen out ODE hours from observational data for the springtime of 23 years (2000-2022).
Then, based on a comparison of the screening results, we investigated the properties of these ODE screening criteria.

### 2.1 Observational Data

Surface ozone mixing ratio and meteorological parameters (pressure, 2-m temperature, 10-m wind speed and direction) at Utqiaġvik (BRW) were taken from the National Oceanic and Atmospheric Administration/Oceanic and Atmospheric Research/Global Monitoring Laboratory (NOAA/OAR/GML) baseline observatory (https://gml.noaa.gov/aftp/data/) (McClure-
Begley et al., 2024; Crocker, 2024), which are freely provided to the public and scientific community. We adopted the data for the years 2000-2022 with a 1-hr time resolution, and only focused on the three months of spring (March, April, and May) in the present study.

### 2.2 Criteria for Identifying ODEs

Three types of criteria (traditional methods, variability-based methods, machine learning methods) were used to screen out
ODE hours from the measurements, which are listed in Table 1 and described in detail below.

#### 2.2.1 Traditional Methods

This kind of methods define ODEs according to the mixing ratios of ozone. The first criterion to be tested is similar to that used by Halfacre et al. (2014), in which ODEs are defined as time periods when ozone mixing ratio falls below 10 ppbv. Moreover,





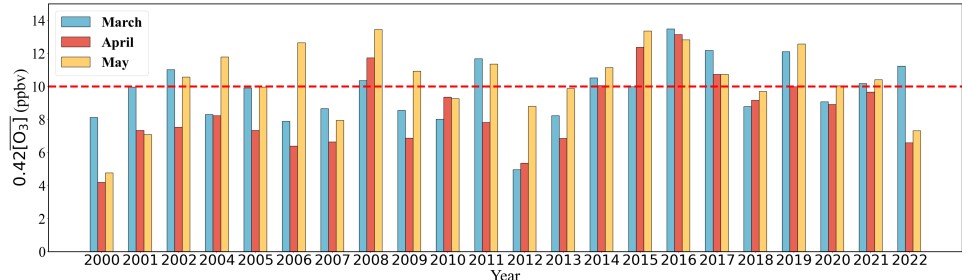

**Figure 1.** Comparison of the value of $0.42\overline{[\mathrm{O_3}]}$ in Eq. (3) against a $10\,\mathrm{ppbv}$ constant (the red dashed line) for the spring months (March, April, and May) from the year 2000 to 2022.

the situation with an ozone value lower than $10\,\mathrm{ppbv}$ should last for longer than one hour (i.e., $[\mathrm{O_3}]{<}10\,\mathrm{ppbv}$ at two consecutive time points). Thus, this criterion can be described as follows:

$$([\mathrm{O_3}]_i < 10) \cap ([\mathrm{O_3}]_{i+1} < 10), \tag{1}$$

in which $[\mathrm{O_3}]_i$ is the ozone mixing ratio at the $i$-th time point. This criterion is referred to as TM1 in the following context (see Table 1). In previous studies (Bottenheim et al., 2009; Frieß et al., 2011; Jacobi et al., 2010; Piot and Von Glasow, 2008; Ridley et al., 2003), different constant thresholds (e.g., 5 ppbv and 4 ppbv) have also been utilized for identifying ODEs. Therefore, in addition to the commonly used $10\,\mathrm{ppbv}$ threshold, we also tested $5\,\mathrm{ppbv}$ and $4\,\mathrm{ppbv}$ thresholds in this study, and named them as TM1-5 ppbv and TM1-4 ppbv (see Table 1).

We then modified TM1 to obtain different screening criteria. At first, we relaxed the duration time of ODEs in the TM1 criterion. Instead of using a consecutive two-hour duration, we defined ODEs as that within the 6 hrs after the first time when ozone is lower than $10\,\mathrm{ppbv}$, there should be at least one hour that ozone is below $10\,\mathrm{ppbv}$ (see TM2 in Table 1):

$$([\mathrm{O_3}]_i < 10) \cap ([\mathrm{O_3}]_{i+j} < 10), j \in \{1,2,3,4,5,6\}. \tag{2}$$

This criterion is able to include time periods when an abrupt increase of ozone occurs during ODEs due to processes such as the stratospheric intrusion and local anthropogenic emissions.

We also replaced the fixed value ($10\,\mathrm{ppbv}$) used in TM1 with a percentage of the monthly averaged ozone value, and named this criterion as TM3:

$$([\mathrm{O_3}]_i < \beta\overline{[\mathrm{O_3}]}) \cap ([\mathrm{O_3}]_{i+1} < \beta\overline{[\mathrm{O_3}]}). \tag{3}$$

In Eq. (3), $\overline{[\mathrm{O_3}]}$ is the averaged ozone value of the corresponding month. $\beta$ is a constant that is artificially given. For a better comparison, we determined the coefficient $\beta$ to be 0.42 using the least squares method, so that $\beta\overline{[\mathrm{O_3}]}$ in Eq. (3) is closest to the constant $10\,\mathrm{ppbv}$ used in TM1 (see Fig. 1).

After that, by integrating TM2 and TM3, we obtained TM4 (see Table 1):

$$([\mathrm{O_3}]_i < \beta\overline{[\mathrm{O_3}]}) \cap ([\mathrm{O_3}]_{i+j} < \beta\overline{[\mathrm{O_3}]}), j \in \{1,2,3,4,5,6\}. \tag{4}$$





Based on TM4, we further took the depleting stage of ozone into consideration. Typically, a complete ODE can be divided into three stages. The first stage is the start of the ODE when the ozone depletes very fast. The second stage is that the ozone remains at a low level (e.g., <10 ppbv), which is the maintenance of ODE. And the third stage is that the ozone returns to background levels, which is the termination of the ODE. The TM4 criterion basically accounts for time periods corresponding

to the second stage. In the next criterion, other than time points that are identified by TM4, time points when the depletion rate of ozone is larger than 1 ppbv h$^{-1}$ were also added,

$$\text{TM4} \cup \left(-\frac{d[\text{O}_3]}{dt} > 1\,\text{ppbv h}^{-1}\right). \tag{5}$$

This criterion further considers time periods representing the first stage of the ODE, and is referred to as TM5, which is listed in Table 1.

### 2.2.2 Variability-Based Method


In our previous work (Cao et al., 2022), we picked out time points representing the tropospheric ODEs at the Halley station in Antarctica according to the variability in the surface ozone mixing ratio, which included considerations of both the mean value and standard deviation. This method for identifying ODEs was proposed based on the study of Bian et al. (2018), when the ozone mixing ratio fulfills the following criterion:

$$[\text{O}_3]_i - \overline{[\text{O}_3]} < \alpha \cdot \sigma, \tag{6}$$

in which $[\text{O}_3]_i$ is the ozone mixing ratio at the $i$-th time point, and $\overline{[\text{O}_3]}$ is the monthly averaged ozone value. $\sigma$ in Eq. (6) denotes the standard deviation. $\alpha$ is a constant which is set to -1.5 in that study (Cao et al., 2022) so that many partial ODEs (i.e., 10 ppbv<[O$_3$]<20 ppbv) can also be identified. According to the criterion described by Eq. (6), ODEs were defined as the time periods when the surface ozone drops to an uncommon low level, rather than the time periods when the surface ozone

falls below a specific threshold, and we named this criterion as VM in the following context (see Table 1).

### 2.2.3 Machine Learning Method

Recently, machine learning methods have been used to investigate ODEs and the associated variability in the tropospheric BrO in the Arctic (Bougoudis et al., 2022). In this study, a machine learning method, Isolation Forest (Liu et al., 2008; Al Farizi et al., 2021), which can detect anomalies in data, was used to screen out ODE hours from the measurements (i.e., IF in Table 1). The

Isolation Forest method is a commonly used machine learning approach for detecting anomalies in data, based on decision tree algorithms. In this method, anomalies are detected by constructing an ensemble of isolation trees, where each tree recursively isolates data points by randomly selecting split values until each point is isolated. After that, the average path length for each point across all trees is calculated to derive an average anomaly score, with higher scores indicating a greater likelihood of being an anomaly. Values of parameters utilized in this method for the current study are outlined as follows: n_estimators (number of

trees), 100; max_samples (number of samples), auto (i.e., using all samples); contamination (proportion of outliers), auto (i.e., auto-select based on data arrangement); max_features (maximum number of features), 1.0 (i.e., using all features).





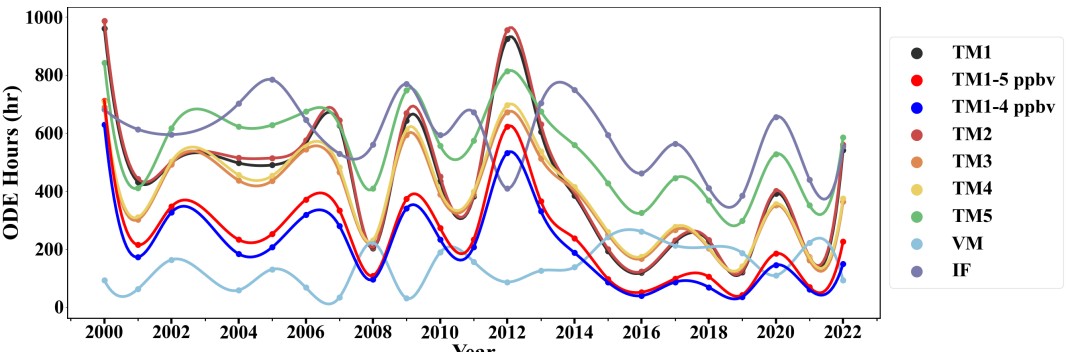

**Figure 2.** Number of ODE hours identified by each criterion from 2000 to 2022.

## 3   Results and Discussions

In this section, we first show the identification results of ODE hours after applying various criteria. These results were also compared and discussed to examine the characteristics and performance of each criterion. Subsequently, we selected several
appropriate criteria for a detailed comparative analysis. This comparative examination allowed us to elucidate the effects of using different criteria on the interannual variability of ODEs and their associations with meteorological parameters.

### 3.1   Interannual Variability of ODE Hours Screened using Various Criteria

Figure 2 depicts the annual changes in ODE hours identified by various criteria across the 2000-2022 period, reflecting the characteristics of each screening criterion. We first focus on the results obtained by the traditional methods (i.e., TM1-TM5).
Generally, the results derived from all the traditional methods display a similar pattern. The years with the most ODE hours were found to be 2000 and 2012. Between the year 2000 and 2012, the ODE hours exhibit a pattern of fluctuation throughout this period. There seems to have an upward trend between 2000 and 2012 if the year 2000 is excluded, suggesting an increase in ODE hours during this time period. This result is consistent with that of Oltmans et al. (2012) reporting a significant increase in the ODE occurrence frequency in the time period of 1973-2010. However, after the year 2012, the curves of all TM methods
show a sharp decline, indicating a significant decrease in ODE hours in recent years. This findings is in accordance with the results of Law et al. (2023) in which they found a pronounced increase trend of the tropospheric ozone at BRW in the springtime of the period 1999-2019.

      With respect to each TM criterion, it can be seen in Fig. 2 that the results using the thresholds of 5 ppbv (TM1-5 ppbv) and 4 ppbv (TM1-4 ppbv) are largely consistent with those using the 10 ppbv threshold (TM1), but with a significant reduction
in the number of ODE hours, because more stringent thresholds were applied. The ODE hours identified using the 5 ppbv and 4 ppbv thresholds are 46% and 55% fewer, respectively, than those identified using the 10 ppbv threshold on average. Figure 2 also shows that results obtained by TM1 and TM2 are almost identical, indicating that influence brought about by the modification of the ODE duration (from a continuous two-hour occurrence to a following one-hour occurrence within a





six-hour period) is negligible. This also means that during occurrences of ODEs, ozone concentrations rarely make a sudden

transition from below 10 ppbv to above 10 ppbv. ODEs typically persist for at least 2 hrs in the majority of cases. Similar

behavior was also found between the results obtained by TM3 (ozone below $0.42\overline{[O_3]}$ for continuous two hours) and TM4

(ozone below $0.42\overline{[O_3]}$ for a start hour and a following hour within a six-hour period). We then focus on the differences

between the results obtained by using TM1 and TM3. It is not surprising to see that when the difference between $0.42\overline{[O_3]}$ and

10 ppbv is significant, there is a considerable discrepancy in the screening results. For instance, the discrepancy in the results

for the year 2012 is remarkable (see Fig. 2), because the mean values of ozone in March and April of this year are quite low

($0.42\overline{[O_3]}$=4.95 and 5.34, respectively); conversely, when the difference is smaller, the discrepancy is less pronounced such as

those for the years 2008 and 2014. As is well known, the background levels of Arctic ozone varies across different years and

seasons, and these changes are associated with factors such as climate change and atmospheric circulation. For instance, the

average ozone levels in the Arctic in the springtime of 2023 were reported to be abnormally higher than historical levels (Ding

et al., 2024). Considering these dramatic changes in the background levels of ozone in the Arctic, using "tailored" thresholds

based on average ozone levels across different time periods (i.e., TM3, TM4) might be another appropriate choice to identify

ODEs. With respect to TM5 (TM4$\cup(-\frac{d[O_3]}{dt}>1\,\mathrm{ppbv\,h^{-1}})$), it is seen in Fig. 2 that the overall trend of the TM5 curve is

generally similar to that of TM4, while the TM5 curve is consistently above the TM4 curve by a margin (152 hrs on average),

indicating the additional hours representing ozone depleting stages considered in the TM5 criterion.

Regarding to the variability-based method (i.e., VM in Fig. 2), its behavior is remarkably different from those TM methods.

Generally, it screened out the least number of ODE hours, indicating it as the most rigorous selection criterion compared with

others. Moreover, it is shown in Fig. 2 that between the year 2000 and 2005, the trend of the VM curve is consistent with

those of TM curves. However, during some other time periods such as from 2010 to 2015, the VM curve displays a trend that

contrasts with the patterns observed in the TM curves. It is because that except the mean value, the VM criterion also takes

the standard deviation into account. When the ozone concentration oscillates greatly, the standard deviation will be high, thus

dominating the criterion given by Eq. (6). As a result, the variability of the criterion would be more closely aligned with the

patterns of the standard deviation, rather than following the mean ozone value. More discussions are given in the following

context.

With respect to the Isolation Forest method (see the IF curve in Fig. 2), generally, the number of ODE hours screened by

this method is comparable to those picked out using TM methods. Interestingly, it is seen that after the year 2014, the IF curve

behaves similarly to those of the TM methods, while before 2014, the IF curve's trend is more like the VM method's trend,

although the values are significantly higher, indicating a possible consideration of the standard deviation in the IF method.

Because of the black box characteristics of machine learning models (Hassija et al., 2024), it is difficult for us to further

explore the reasons and principles behind the screening results of this method. Further interpretability of this machine learning

method is also one of areas we aim to investigate in the future.

In the subsequent analysis, we will focus on two specific years (2012 and 2021) to investigate more deeply into the charac-

teristics of these criteria for identifying ODEs. These two years were selected because 2012 is one of years with the most ODE





hours (530-950 hrs indicated by the TM methods). In contrast, 2021 is one of years with the least ODE hours (60-350 hrs), but also having a frequent oscillation in ozone levels.

## 3.2 ODE Hours on Specific Years Identified by Different Criteria

Because the results of TM1 are almost identical to those of TM2, and the results of TM3 are largely in line with TM4. Additionally, results of TM1-5 ppbv and TM1-4 ppbv show a decrement from TM1's results, and results of TM5 show an increment from TM4's, while maintaining essentially a similar trend. Therefore, in this section, we only compare the results of ODE hours screened by TM1, TM4, VM, and IF methods on specific years (i.e., 2012 and 2021).

### 3.2.1 Traditional Methods

Figure 3(a) and (b) illustrate the results of screened ODE hours using the TM1 criterion for the years 2012 and 2021. We found that in the year 2012, because the average ozone level is lower than that in 2021, the identified ODE hours in 2012 (i.e., 925 hrs) are significantly more than those in 2021 (i.e., 165 hrs). This implies that when TM1 is applied, the number of identified ODE hours is positively correlated with the annual average ozone concentration for a specific year, which can be easily expected since a year with a lower ozone level is more likely to meet the fixed threshold criteria.

The TM1 criterion is straightforward and easy to apply. However, Fig. 3(b) shows that in 2021, many time points when the ozone mixing ratio decreased sharply were not identified as ODE hours because the depletion is not strong enough to reduce the ozone level to below 10 ppbv. This is also the possible reason why another criterion for identifying "partial" ODEs (i.e., 10 ppbv<$[O_3]$<20 ppbv) is often suggested in many previous studies (Ridley et al., 2003; Koo et al., 2012; Halfacre et al., 2014).

In contrast to the TM1 method, TM4 gives different thresholds according to the monthly averaged ozone value across different months and years. Generally, the results obtained by TM4 are consistent with those obtained by TM1 (see Fig. 3c and d). In months and years that possess low ozone concentrations (<15 ppbv, e.g., year 2012, see Fig. 3c), the implementation of TM4 would exert a more stringent threshold so that less ODE hours are identified. In contrast, in month and years that posses high ozone concentrations (e.g., year 2021, see Fig. 3d), comparable or more ODE hours can be identified using this criterion.

### 3.2.2 Variability-Based Method

As discussed above, the variability-based method screened out significantly fewer ODE hours than the other methods. For instance, in 2012, the VM method identified less than 200 ODE hours, whereas the other methods each screened more than 500 hrs. Therefore, we focus on specific years to clarify the reasons. Figure 3(e) shows that for the year 2012, the VM method identified only a few ODE hours from the time series of ozone. Moreover, these identified ODE hours all reside in the month of May, whereas no ODE hours were identified in March and April of 2012. Similar results were also found in some other years such as 2013 and 2022 (not shown here). The reason for this ODE underestimation is that when the monthly averaged ozone level is low, being below 1.5 times the standard deviation, a negative threshold would be calculated using Eq. (6), which cannot







**Figure 3.** Screened results for the years 2012 and 2021 using various criteria. The blue curve represents the hourly time series of the ozone mixing ratio, and the red dots denote the ODE hours identified by various criteria.

be fulfilled. As a result, the VM method would give a null ODE identification for that month. In contrast, the ODE hours in

2021 screened by the VM method are more reasonable, shown in Fig. 3(f). This is because the monthly averaged ozone levels for this year fall within a typical range ($\sim$20-30 ppbv), which is moderately greater than 1.5 times the standard deviation ($\sim$10-18 ppbv). Consequently, a more reasonable criterion is derived from Eq. (6). Thus, when using the VM method to determine



ODE hours, extra caution is required for months characterized by notably low average ozone levels and pronounced oscillations in the mixing ratio.

To solve the problem of this ODE underestimation, we relaxed the constant $\alpha$ in Eq. (6) from the original value of -1.5 designed for Halley Station to a value of -0.8. Consequently, more reasonable results were obtained for the year 2012 (see Fig. S1a in the Supplementary Information). The interannual variability of ODE hours determined by this modified criterion is also more consistent with those determined by other criteria (see Fig. S2 in the Supplementary Information). However, in years and months with high ozone levels, this modified criterion with a smaller $\alpha$ seems to give an excessively high number

of ODE hours (see Fig. S1b in the Supplementary Information), which is also not that appropriate. We also tested other values of $\alpha$ such as 1.0, but still did not obtain satisfactory screening results for the ODEs in 2012 at BRW (not shown here). Thus, we concluded that for the variability-based method, the original value of $\alpha$ (i.e., -1.5) designed for Halley Station in Antarctica may not be suitable for identifying ODEs in certain years and months at other stations, and a more appropriate value for $\alpha$ should be carefully determined for different years, months, and stations.

### 3.2.3   Machine Learning Method

Results of the IF method are very interesting. For years with normal ozone levels such as 2021 (see Fig. 3h), the IF method performs well, identifying not only hours with low ozone but also hours when ozone suddenly drops. However, for years with relatively low ozone levels such as 2012 (see Fig. 3g), the IF method gives problematic results. It was found in Fig. 3(g) that the hours with a moderately low ozone value (3-7 ppbv) were not recognized as ODE hours. The reason for this is that

during the training process of the machine learning model, dense data were regarded as normal whereas sparse data were considered as outliers. Thus, for years with only a few ODE occurrences such as 2021, the IF method is capable of identifying ODE hours by recognizing them as outliers. However, in years with a frequent occurrence of ODEs such as 2012, hours with moderately low ozone values are regarded as normal so that they are not viewed as ODE hours by this model. Thus, the machine learning approach exhibits a limitation in accurately identifying ODE hours in years characterized by a high frequency of ODE

occurrences.

From the results discussed above, we found that TM1 and TM4 are more suitable for identifying ODEs from the time series of ozone at the BRW station than the other criteria. We then investigated the monthly and yearly variability of ODE hours at the BRW station based on the results applying these two criteria. Another two criteria with different constant thresholds (i.e., TM1-5 ppbv and TM1-4 ppbv) were also examined to assess the impact of varying the constant threshold on the conclusions.

### 3.3   Variability of ODE Hours at BRW

We first analyzed the monthly variation in ODE hours across various spring months over these 23 years (see Fig. 4). It is shown in Fig. 4(a) that when the constant 10 ppbv criterion was applied, April has the most ODE hours, with a median of approximately 210 hrs, suggesting that April is the predominant month when ODEs occur. In contrast, May possesses the least ODE hours, with a median of 57 hrs. Moreover, Fig. 4(a) shows that in March, the number of ODE hours ranges from zero to

more than 400 hrs, with the majority concentrated around 170 hrs. In contrast, the range of ODE hours for April is narrower,







**Figure 4.** Monthly hours of ODEs for March, April, and May across the 23-year period from 2000 to 2022. The ODE hours were screened by (a) TM1, (b) TM4, (d) TM1-5 ppbv and (d) TM1-4 ppbv.

spanning from 50 to 400 hrs, but with a more uniform distribution of hours throughout this range. It means that the ozone concentration at the BRW station fluctuates more widely in March. For May, this month possesses a relatively small range of ODE hours (0-350 hrs), with the bulk of hours centered at a low value (∼50 hrs).

When the TM4 criterion (ozone below $0.42\overline{[O_3]}$ for a start hour and a following hour within a six-hour period) was applied (Fig. 4b), we found that the order of the medians for these three months remains unchanged (April>March>May), compared with the results of TM1. However, we found that the ODE hours in April lowered while the hours in May elevated, when TM4 was applied. The data are also more concentrated. It is because that TM4 uses a threshold that depends on the monthly averaged value. As the average ozone level in April is lower compared with those in March and May, using this relative criterion would exert a more stringent threshold than the constant 10 ppbv so that it screens out less ODE hours than TM1. For May with a high



ozone level, the relative criterion is easier to achieve compared to the constant 10 ppbv criterion. Therefore, TM4 identifies more ODE hours in May compared to TM1. In summary, the TM4 method can screen out more ODE hours in months with high ozone values than TM1, and vice versa.

When 5 ppbv was used to replace the 10 ppbv threshold (i.e., TM1-5 ppbv, see Fig. 4c), we found that the order of ODE hours across the three spring months remained unchanged (April>March>May). However, the ODE hours for each month

were significantly lower compared to those identified by TM1, due to the stricter threshold applied. Furthermore, the discrepancy in ODE hours between March and April was found to be less pronounced than that in the TM1 results, rendering the ODE hours in March (median: 87 hrs) and April (median: 102 hrs) very close. This indicates that the primary cause for the significantly greater number of ODE hours in April compared to March, as identified by TM1, is the more frequent occurrence of ozone concentrations falling within the 5-10 ppbv range in April than in March. With respect to the situation in May, the

implementation of the 5 ppbv threshold resulted in a significant reduction to almost zero ODE hours. This finding suggests that ODEs characterized by very low ozone levels (<5 ppbv) have nearly vanished in May in recent years, which could be linked to global warming, resulting in higher temperatures during May, thereby causing the ODE season to cease earlier than before (Burd et al., 2017). Results obtained using the 4 ppbv threshold (Fig. 4d) are similar to those using the 5 ppbv threshold, except that the discrepancy in ODE hours between March (median: 73 hrs) and April (median: 85 hrs) is even less pronounced, thereby

confirming the conclusions drawn above.

Figure 5 presents the yearly variability of the ODE hours for each spring month and the entire spring season, spanning from 2000 to 2022. It is seen from Fig. 5(a) that based on the results of TM1, ODE hours in spring decrease significantly across these 23 years (p<0.05). This decreasing trend is mostly caused by the decline in ODE hours in April (see Fig. 5e), which is highly significant (p<0.01). In contrast to April, the drops in ODE hours in March and May are not significant (p>0.1, Fig. 5c)

and close to significant (0.05<p<0.1, Fig. 5g), respectively. Our findings are in good agreement with those of Law et al. (2023), who reported a notable increase in observed surface ozone levels during spring from 1993 to 2019 at BRW, with the most significant elevation observed in April.

The results of TM4 are similar (Fig. 5b), but showing a more significant decline trend. The decrease in ODE hours in the entire spring season transits to be highly significant (p<0.01) in the results of TM4, and the p-value in April is also smaller,

confirming the highly significant decline in ODE hours in April at BRW over the 23-year period. Aside from that, the drop in ODE hours in May was found to be insignificant (p>0.1) in TM4's results, whereas in TM1's results, the drop in May approached statistical significance (0.05<p<0.1).

When more rigorous thresholds (5 ppbv and 4 ppbv) were applied instead of the 10 ppbv threshold (refer to Fig. S3 in the Supplementary Information), the reduction in ODE hours in the entire spring season was found to be highly significant

(p<0.01), denoting a more remarkable decline in the occurrence of ODEs with very low ozone levels during these years. This remarkable decline is still mainly attributable to the highly significant reduction in ODE hours in April (p<0.01). The decrease in March remains statistically insignificant, while the decline in May is identified as insignificant (p>0.1) for the 5 ppbv threshold and close to significant (0.05<p<0.1) for the 4 ppbv threshold, respectively.





**Figure 5.** Yearly variability of the ODE hours at the BRW station, identified by two different criteria. Subplots (a), (c), (e) and (g) show the ODE hours screened by the TM1 method for the whole spring, March, April and May, respectively, and subplots (b), (d), (f) and (h) show the hours screened by the TM4 method. Red dashed lines represent linear regressions of the ODE hours. The regression equations and p-values are also given.





In summary, the results utilizing different criteria all indicate a decline in ODE hours during the spring season over the 23-
year period, primarily driven by the highly significant reduction in April. However, when employing the threshold that varies
with the monthly average $(0.42\overline{[O_3]})$ or more stringent thresholds (5 ppbv and 4 ppb), the results would show a more significant
decline in ODE hours in spring, compared to that using the 10 ppb threshold.

### 3.4  Relationship between ODE Hours and Meteorological Parameters

We then investigated the relationship between ODE hours and surface meteorological parameters measured at the BRW sta-
tion. Figure 6 shows the wind information during the investigated spring seasons (Fig. 6a) and the time periods of ODEs
identified by TM1 (Fig. 6b) and TM4 (Fig. 6c). From Fig. 6(a), we can see that the prevailing wind at BRW is northeastern
in the springtime. The wind speeds are relatively higher when the winds are easterly (9-12 m s$^{-1}$ at most) and northeasterly
(>12 m s$^{-1}$ at most). In contrast, wind speeds along other directions are mostly lower than 9 m s$^{-1}$. Compared to the wind
speeds throughout the whole spring, Fig. 6(b) shows that during ODEs, the wind speeds are significantly lower (below 9 m s$^{-1}$,
mostly 3-6 m s$^{-1}$), denoting a favored moderate wind speed condition of ODEs at BRW. Moreover, the wind direction tends
to favor the northeasterly and northerly winds during ODEs, which is associated with the fresh sea ice covering the ocean to
the north of BRW (Bottenheim and Chan, 2006; Gilman et al., 2010; Oltmans et al., 2012; Peterson et al., 2016). Additionally,
we also found an increased proportion of westerly winds and a decreased proportion of easterly winds in Fig. 6(b), denoting a
favored westerly wind condition for ODEs at BRW. In an earlier study of Oltmans et al. (2012), they suggested that ODEs at
BRW are predominantly associated with easterly winds, a finding that contrasts with the conclusions achieved in this study. The
discrepancy between our findings and those of Oltmans et al. (2012) may source from the different time periods investigated
in these two studies. Moreover, a recent flight campaign originating from BRW (Brockway et al., 2024) also detected elevated
levels of BrO being advected from the west of BRW, suggesting a potential transport of ozone-depleting air from the western
direction. Additionally, in our recent modeling study on ODEs at BRW (Cao et al., 2023), we also found an ozone-depleting
air mass transported to the BRW station from the southwest under the influence of a cyclone moving eastward. The results
obtained by applying the TM4 criterion (Fig. 6c) are also similar, confirming our findings.

Results utilizing the 5 ppbv and 4 ppbv thresholds (refer to Fig. S4 in the Supplementary Information) also suggested that
northerly and northeasterly wind conditions would facilitate the occurrence of ODEs at BRW. However, relative fractions of
the moderate wind speed (3-6 m s$^{-1}$) and the low wind speed (0-3 m s$^{-1}$) during the identified ODE periods are higher than
those found in the TM1's results, indicating that lower wind speeds are conducive to the occurrence of more severe ODEs,
characterized by very low ozone levels.

The connections between ODE hours and the 2-m temperature measured at BRW are shown in Fig. 7, and the results obtained
by applying TM1 (Fig. 7a) and TM4 (Fig. 7b) are similar. It was found that in the range of 250-272 K, the temperature at BRW
generally exhibits a uniform distribution. However, occurrences of ODEs are more frequent at a lower temperature, with the
highest occurrence frequency at approximately 250 K. Moreover, when the temperature is lower than approximately 256 K, the
decrease in the 2-m temperature would substantially enhance the occurrence of ODEs (see the blue dot-dash lines in Fig. 7a and
b). Thus, a lower temperature condition can facilitate the occurrence of ODEs, which is possibly associated with the stability



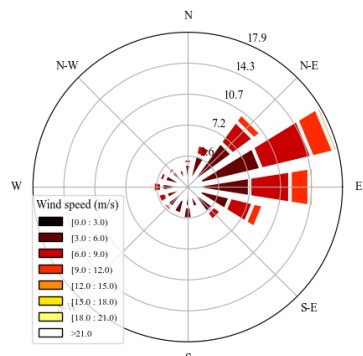

(a) Spring

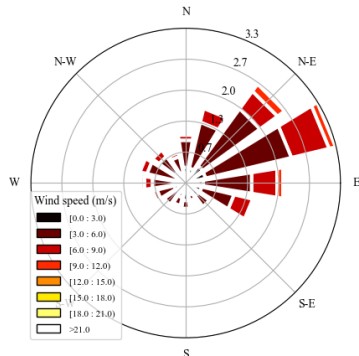

(b) ODE hours (TM1)

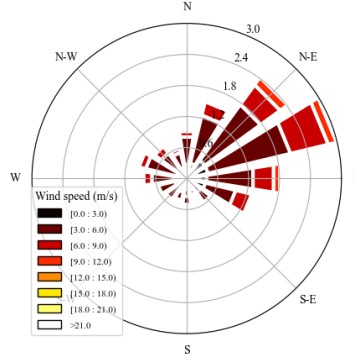

(c) ODE hours (TM4)

**Figure 6.** Wind rose diagrams during (a) the investigated spring seasons from 2000 to 2022, and ODE time periods identified by (b) TM1 and (c) TM4.



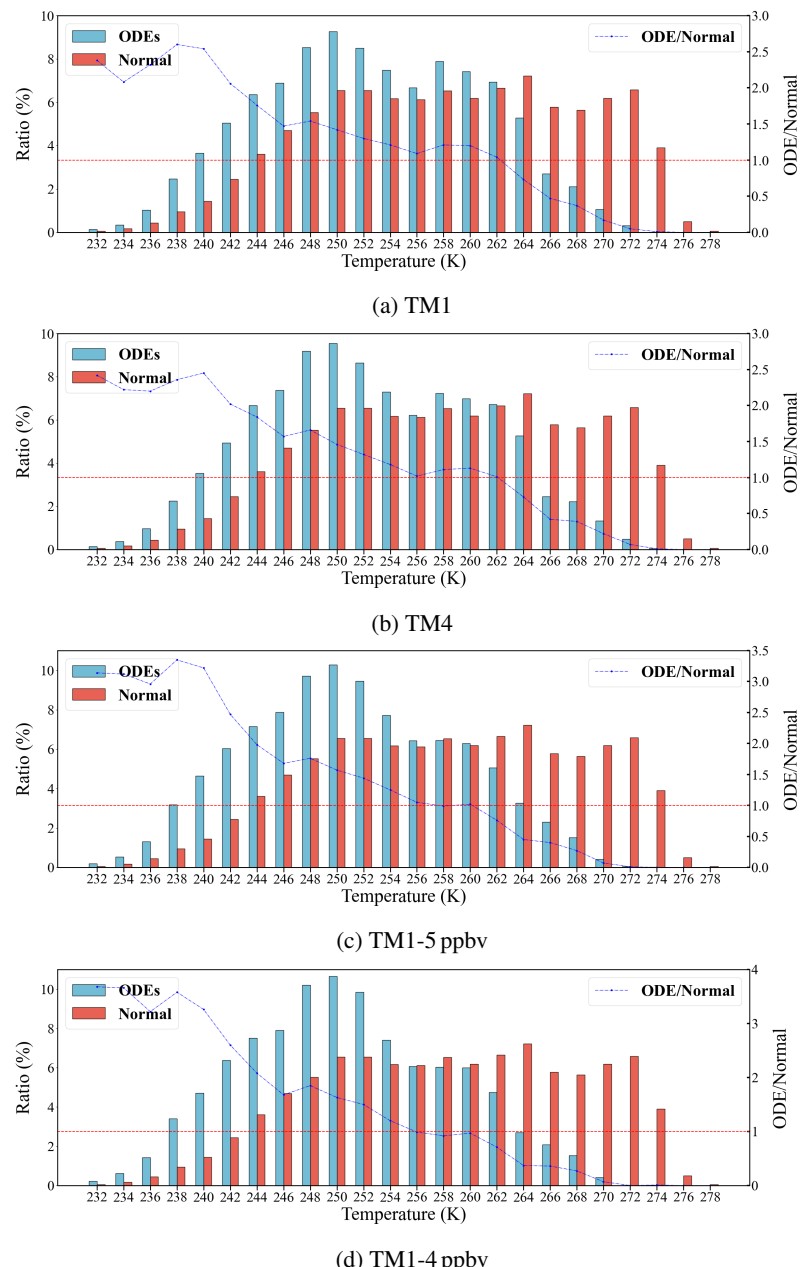

(a) TM1

(b) TM4

(c) TM1-5 ppbv

(d) TM1-4 ppbv

**Figure 7.** Occurrence frequency of ODEs versus the 2-m temperature. The ratio was calculated by the number of hours within each interval divided by the total number of hours. ODE hours used to calculate the ratio in (a), (b), (c) and (d) were screened by TM1, TM4, TM1-5 ppbv and TM1-4 ppbv, respectively. The blue dot-dash lines denote the ratio of ODEs to normal conditions, and a value greater than 1.0 indicates a favorable condition for ODEs.



of the boundary layer (Lehrer et al., 2004; Koo et al., 2012). It could also be linked to the effective absorption of HOBr on frozen NaCl/NaBr surfaces at temperatures below the eutectic point of NaCl·2H2O (i.e., 252 K), as reported by Adams et al.

(2002). Below this temperature, a quasi-brine layer, characterized by its high acidity (Cho et al., 2002), is likely to develop on the ice/snow surface, which would promote bromine activation and subsequent ozone depletion. This lower temperature condition favored by the bromine explosion and the ozone depletion was also reported by Zilker et al. (2023), by applying a composite analysis on long-term ozonesonde data (2010-2021) and surface measurements over the Svalbard area in the Arctic.

Results obtained by applying TM1-5 ppbv and TM1-4 ppbv (Fig. 7c and d) also indicate that ODEs are favored by low

temperature conditions. More interestingly, Fig. 7(c) and (d) reveal a significant reduction in the occurrence frequency of severe ODEs within the temperature range of 256-262 K, when compared to the TM1's results (Fig. 7a). This suggests that severe ODEs are more likely to occur only at temperatures below 256 K.

Figure 8 show the relationship between the surface pressure of BRW and the occurrence of ODEs. Results obtained by using various criteria are largely consistent, exhibiting only slight differences. We found the occurrence of ODEs mostly residing

in the pressure range of 1012-1027 hPa. Furthermore, an increase in surface pressure at BRW is often associated with a more frequent occurrence of ODEs, possibly due to the stable and calm conditions that prevail under the control of high-pressure systems. Additionally, it can be seen in Fig. 8 that when the surface pressure resides in the range of 990-995 hPa, the occurrence of ODEs is also favored (with the ratio ODE/Normal >1.0). It might be connected to the blowing snow events (Yang et al., 2010, 2019, 2020; Huang et al., 2018, 2020) when low-pressure systems (e.g., cyclones) pass by, which are beneficial for the

BrO release and the subsequent ozone depletion (Begoin et al., 2010; Zhao et al., 2016).

## 4 Conclusions and Future Work

In this study, we investigated the influence of applying various criteria to identify springtime tropospheric ozone depletion events (ODEs) at Utqiaġvik (BRW), Arctic, using observational data from 2000 to 2022. We tested three types of criteria: traditional methods, variability-based methods, and machine learning methods, and analyzed the characteristics of these criteria

in depth.

We found the criteria using a constant threshold (e.g., 10 ppbv) and using thresholds based on the monthly averaged ozone values more suitable for identifying ODEs at BRW than the other criteria. Furthermore, all these different criteria indicate an overall decreasing trend in the occurrence frequency of ODEs at BRW over this 23-year period, with the most significant decline observed in April. This suggests a potential impact of climate change such as the global warming and Arctic sea ice

melting on ODE occurrences. However, results obtained by implementing a threshold that varies with the monthly average, or by applying more stringent thresholds (5 ppbv and 4 ppbv), demonstrate a more significant reduction in the occurrence of ODEs compared to those using the 10 ppbv threshold.

Applying suitable criteria also enables us to study the connection between meteorological conditions at BRW and the occurrence of ODEs more precisely. ODEs at BRW were found to be more likely to occur under northerly and northeasterly winds, with moderate wind speeds (mostly 3-6 m s$^{-1}$) being more favorable, although ODEs were also observed at higher





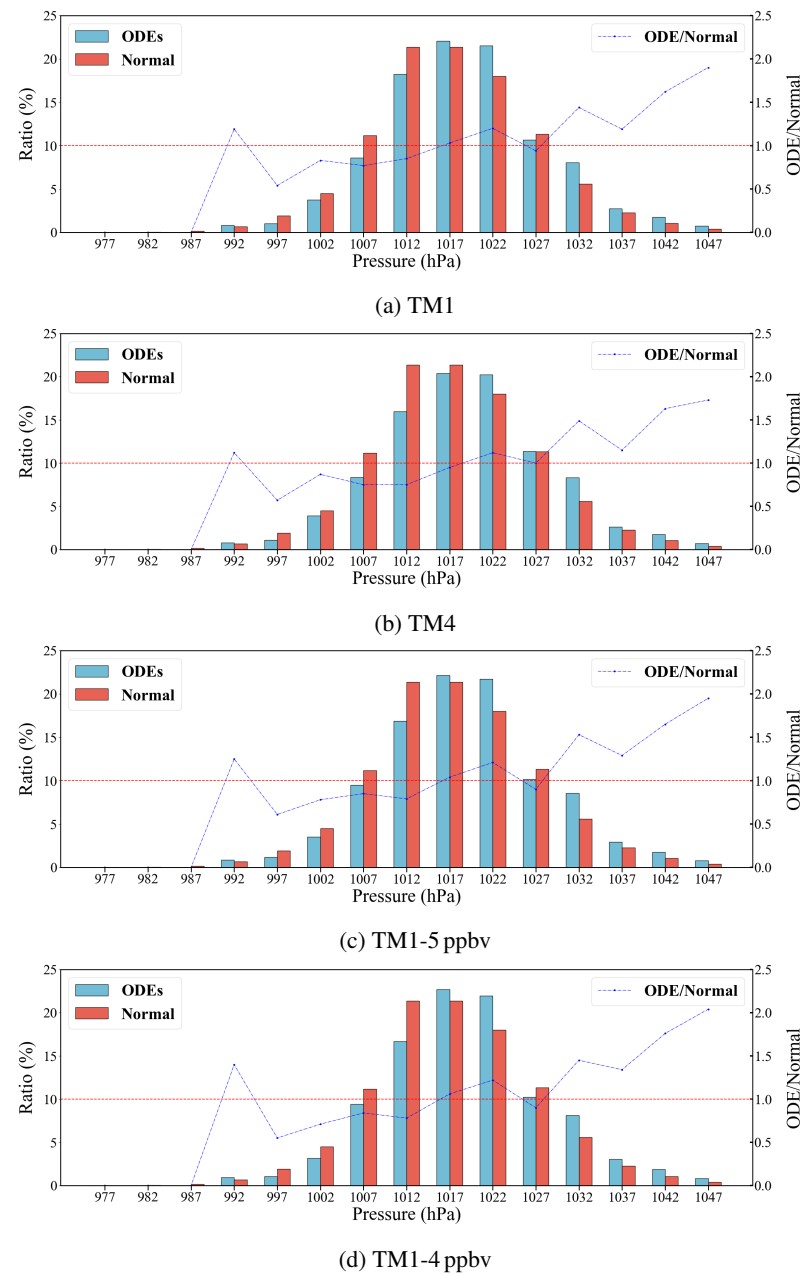

**Figure 8.** Occurrence frequency of ODEs versus the surface pressure. The other settings for this figure are the same as those for Fig. 7.

wind speeds (6-12 m s$^{-1}$). Lower wind speed conditions were also found to facilitate the occurrence of more severe ODEs, characterized by very low ozone concentrations. ODEs were also found to be closely linked to temperature and pressure. ODEs especially the severe ones tend to occur at temperatures lower than 256 K, which is possibly associated with the stability of



the boundary layer and the effective absorption of HOBr on frozen surfaces, promoting bromine activation and subsequent
ozone depletion. Additionally, ODEs at BRW tend to occur under high pressure conditions (>1010 hPa), indicating that the
high-pressure associated weather conditions may facilitate the occurrence of ODEs.

In the future, we would like to improve the machine learning methods (e.g., Isolation Forest) so that a more reliable method
can be applied in ODE identification. Moreover, present analysis can also be conducted to a longer time series of data to
provide a more comprehensive understanding of the long-term trends and variability of ODEs in the Arctic. In addition,
other observational data, such as BrO measurements from satellite detection and ground-based multi-axis differential optical
absorption spectroscopy (MAX-DOAS), can also help to identify ODEs more accurately.

*Code and data availability.*  All data needed to evaluate the conclusions in the paper are present in the paper. The source code of the model
and the data of the computational results shown in this article can be acquired upon request from the authors.

*Author contributions.*  L.C. conceptualized the study and supervised the entire research process. X.Z. conducted the simulations and pro-
cessed the data. X.Y. and S.L. contributed to the interpretation of the results. J.W. provided valuable insights into the model results. T.Z.
assisted with the data analysis and manuscript preparation. All authors discussed the results and contributed to the final manuscript.

*Competing interests.*  The authors declare no conflict of interest.

*Acknowledgements.*  This study is funded by the National Key Research and Development Program of China (Grant No. 2022YFC3701204),
the National Natural Science Foundation of China (Grant No. 41705103), and the 2023 Outstanding Young Backbone Teacher of Jiangsu
''Qinglan'' Project (Grant No. R2023Q02). The authors would like to thank the National Supercomputer Center in Tianjin and High Per-
formance Computing Center at the Nanjing University of Information Science and Technology to provide the high-performance computing
system for calculations.



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
