# Peer review of "Influence of Various Criteria on Identifying the Springtime Tropospheric Ozone Depletion Events (ODEs) at Utqiagʻvik, Arctic"

_EGUsphere, 2024_

## Author Comment (AC1)

**Reply To Referee #1**

We would like to thank Referee #1 for the professional revision on our manuscript. We have modified our manuscript according to the suggestions. In the following, we made the point-to-point reply to the comments raised by Referee #1.

**Q1.1 The paper is quite complete. However, some minor changes should be considered prior to its final acceptance.**

**A1.1** Thanks a lot for the positive evaluation on our manuscript and valuable suggestions that significantly improved our paper.

**Q1.2 Since the analysis is made on a specific site, the interested readers could wonder about the result robustness, i.e., the authors should comment about limitations of the result extension to different sites.**

**A1.2** Thanks a lot for raising the issue regarding the robustness and limitations of our conclusions. We fully understood the potential concerns of Referee #1 about extending our findings to different observation sites. Therefore, during the revision process, we gathered more observational data from 7 other Arctic sites (Alert, Esrange, Tustervatn, Villum, Pallas, Summit, and Zeppelin) aside from the one (Barrow) we initially focused on. After applying the criteria we proposed in the present study (10ppb and 5ppb), only four of these sites (Alert, Barrow, Villum, and Zeppelin) were found to have ODEs (Ozone Depletion Events) occurrences (see Fig. A1 in this rebuttal). It should be noted that the availability and the quality of the observational data vary significantly among these sites. For instance, the Alert site only has data spanning from 2000 to 2012, which amounts to 13 years. In contrast, the Villum site possesses a dataset with a longer time range, covering a time period from 2008 to 2022, totaling 15 years.

[Figure]

[Figure]

Figure A1 Occurrence frequency of ODEs at different stations (Alert, Utqiagvik, Villum, and Zeppelin). The occurrence frequency is calculated as the ratio of ODE hours to total hours, in which the ODE hours are identified using the TM1 and TM1-5 ppb screening criteria.

From the results shown in Fig. A1 in this rebuttal, we found that the ODE hours picked out by the criteria proposed in this study seem appropriate. For instance, Barrow, which has a low altitude, is featured with high frequency of ODE occurrence. In contrast, Zeppelin, which is located at a higher altitude, has fewer ODE hours, because the air mass arriving at Zeppelin usually represents the air in the free troposphere so that more ozone can be transported from the stratosphere and less halogens released from the surface can reach the free troposphere due to the barrier at the top of the boundary layer. Additionally, the ODE curves for Villum and Barrow (see Fig. A1 in this rebuttal) both show a declining trend, which is consistent with the conclusion achieved in this study. However, more observational data for other species such as halogen species (i.e., BrO) are still needed to validate these results, which is also a limitation of the present study.

Because the ODE occurrence at different stations depend on many factors such as the altitude and the location of the station, and a thorough comparison of the results and a detailed discussion on this topic can form another interesting paper, at present we only discussed the screened results briefly in the conclusion section and attributed this work to a future publication. We appreciate the valuable suggestion from Referee #1. The added discussion can be found in lines 441-454 in the revised manuscript and Fig. S8 in the revised Supplementary Information.

**Q1.3 In paragraph between lines 245 and 254, the authors discuss about the suitable number of ODE hours. They should indicate a reason for such suitable number or when a number could be excessive.**

**A1.3** Thank you for the comment regarding the suitable number of ODE hours in our manuscript. In the case shown in the present manuscript, when we set the parameter $\alpha$ in Eqn. (6) to $-0.8$, this criterion can identify a significantly high number of ODE hours. Specifically, for the year 2021 (see Fig. A2 in this rebuttal), this method recognized many data points with ozone mixing ratios between 15 and 20 ppb, which are often referred to as "partial ozone depletion" events in previous studies (Ridley et al., 2003; Piot and von Glasow, 2008). However, the springtime average ozone mixing ratio at BRW for the year 2021 was calculated to be 23.93 ppb. It means that many time points picked up by this criterion possess an ozone value close to the average ozone level in

springtime at BRW. In that case, we feel that many of these time points cannot be viewed as ODE hours, which also indicates that the criterion overestimates the ODE hours. This overestimation in ODE hours may also lead to a misunderstanding of the trend of ODE occurrence.

[Figure]

Figure A2. Screened results for the year 2021 using the modified VM criteria, in which α is set to -0.8. The blue curve represents the hourly time series of the ozone mixing ratio, and the red dots denote the ODE hours identified by the criterion.

Please see lines 275-281 in the revised manuscript for explanations added to the manuscript to clarify this issue.

**Q1.4 Finally, the references used for discussion are unevenly distributed. For instance, they are frequent for the relationship with meteorological parameters, but less frequent in previous result sections. References should be the link between this research and previous studies.**

**A1.4** Thanks for your suggestions. In fact, our findings about the trend of ODE occurrences are in good agreement with those of Law et al. (2023), who reported a notable increase in observed surface ozone levels in springtime especially in April of BRW from 1993 to 2019. Hung et al. (2025) also observed an increasing trend in Arctic spring ozone concentrations at Eureka, Nunavut, Canada (80°N, 86°W) from 2008 to 2022, further supporting the notion of declining ODE frequency. Burd et al. (2017), in their study on the Arctic BrO season, found a decrease in BrO concentrations and an early end of the BrO season at BRW from 2012 to 2016, which may also imply a reduction in the occurrence of ODEs, aligning with our findings.

We have added these references and related discussions into the parts about trend of ODE occurrences in the revised manuscript. Please see lines 339-343 in the revised manuscript.

**Q1.5 Advantages and disadvantages of the presented procedures should be highlighted in the conclusions.**

**A1.5** Thanks a lot for your comment. We added a discussion about the advantages and disadvantages of the present study in the conclusion section as suggested. The added content is as

follows:

"We found the criteria using a constant threshold (e.g., 10 ppbv) and using thresholds based on the monthly averaged ozone values more suitable for identifying ODEs at BRW than the other criteria. In contrast, the criterion considering both the mean value and standard deviation of ozone (i.e., the VM criterion) is able to identify time points when the surface ozone drops to an uncommon low level instead of a fixed threshold, which is more adaptive and sensitive. However, extra caution is required when determining the parameter value of this criterion. Apart from these criteria, the machine learning method adopted in this study (i.e., the IF method) can automatically detect ODE hours. But this method has a poor interpretability in screening results and sometimes is unable to correctly identify ODE hours when ODEs occur very frequently."

Please also see lines 412-419 in the revised manuscript.

**Minor remarks.**
**Q1.6 Line 46. Replace "1999),." By "1999)."**
**A1.6** Modified. Thanks.

**Q1.7 Since curves are superposed in Fig. 2. Perhaps, additional information could be obtained if each criterion is represented by its mean and standard deviation and all of them are ordered following their means.**
**A1.7** Thank you very much for the suggestion. We have modified Figure 2 in the manuscript according to the suggestion (see Fig. A3 in this rebuttal). At present, the figure is divided into two subplots. The upper subplot presents the results of the TM series methods, while the lower subplot shows the results of the VM and IF methods. Furthermore, we have re-ordered the curves in each subplot according to the mean values of the criteria. Additionally, mean values and standard deviations are also added into the figure according to the reviewer's suggestion. We believe that these modifications will provide additional information and make the figure more informative and easier to understand. Thanks again for your valuable suggestion.

[Figure]

Figure A3 Number of ODE hours identified by each criterion from 2000 to 2022.

**References**

Burd, J. A., Peterson, P. K., Nghiem, S. V., Perovich, D. K., and Simpson, W. R.: Snowmelt onset hinders bromine monoxide heterogeneous recycling in the Arctic, Journal of Geophysical Research: Atmospheres, 122, 8297–8309,https://doi.org/https://doi.org/10.1002/2017JD026906, 2017.

Hung, J., Liu, L., Palm, M., Mariani, Z., Manney, G. L., Millán, L. F., and Strong, K.: Autonomous year-round measurements of $O_3$, CO, $CH_4$, and $N_2O$ in the High Arctic with the Atmospheric Emitted Radiance Interferometer, Journal of Geophysical Research: Atmospheres, 130, e2024JD042847, https://doi.org/10.1029/2024JD042847, 2025.

Law, K. S., Hjorth, J. L., Pernov, J. B., Whaley, C., Skov, H., Collaud Coen, M., Langner, J., Arnold, S. R., Tarasick, D. W., Christensen, J., Deushi, M., Effertz, P., Faluvegi, G., Gauss, M., Im, U., Oshima, N., Petropavlovskikh, I., Plummer, D., Tsigaridis, K.,Tsyro, S., Solberg, S., and Turnock, S. T.: Arctic tropospheric ozone trends, Geophysical Research Letters, 50, e2023GL103 096,510,https://doi.org/10.1029/2023GL103096, 2023.

Piot, M. and Von Glasow, R.: The potential importance of frost flowers, recycling on snow, and open leads for ozone depletion events, Atmospheric Chemistry and Physics, 8, 2437–2467, 2008.

Ridley, B. A., Atlas, E. L., Montzka, D. D., Browell, E. V., Cantrell, C. A., Blake, D. R., Blake, N. J., Cinquini, L., Coffey, M. T., Emmons, L. K., Cohen, R. C., DeYoung, R. J., Dibb, J. E., Eisele, F. L., Flocke, F. M., Fried, A., Grahek, F. E., Grant, W. B., Hair, J. W., Hannigan, J. W., Heikes, B. J., Lefer, B. L., Mauldin, R. L., Moody, J. L., Shetter, R. E., Snow, J. A., Talbot, R. W., Thornton, J. A., Walega, J. G., Weinheimer, A. J., Wert, B. P., and Wimmers, A. J.: Ozone depletion events observed in the high latitude surface layer during the TOPSE aircraft program, Journal of Geophysical Research: Atmospheres, 108, TOP 4–1–TOP 4–22, https://doi.org/https://doi.org/10.1029/2001JD001507, 2003.

---

## Author Comment (AC2)

**Reply To Referee #2**

We would like to thank Referee #2 for the valuable comments on our manuscript, which greatly improved our manuscript. We have revised our manuscript according to the suggestions. In the following, we made the point-to-point reply to the comments raised by Referee #2.

**Major comments:**

**Q2.1 The discussion is a bit superficial. Especially in the subsequent impact of different criteria on ODEs event identification (i.e., health impact, climate change, etc.). Besides, as mentioned in the final "Conclusions and Future Work", authors should involve more observation data (such as BrO) to further evidence the assumption or the attribution in the discussion.**

**A2.1** Thanks a lot for your insightful comments on our manuscript.

First, about the discussion on the subsequent impact of using different criteria to identify ODEs, in this study, we found that when using a criterion based on a constant threshold (10 ppbv or 5 ppbv) or thresholds considering the monthly averaged ozone and the standard deviation, an overall decline in the occurrence frequency of ODEs at Utqiagvik, Arctic from the year 2000 to 2023 can be revealed. Our findings are also in good agreement with those of Law et al. (2023), who reported a notable increase in observed surface ozone levels in springtime especially in April of BRW from 1993 to 2019. Burd et al. (2017), in their study on the Arctic BrO season, found a decrease in BrO concentrations in springtime and an earlier ending of the BrO season at BRW from 2012 to 2016, which may also indicate a reduction in the occurrence of ODEs at BRW, aligning with our findings. This declining trend of ODEs can lead to a weakening of the deposition of active mercury (Hg(II)), which is highly toxic and can pose serious health risks to humans. Therefore, the decline in ODE frequency could lead to a reduction in the health hazards associated with mercury deposition in mid-latitude regions. However, when a different criterion is used, a more significant declining trend of ODEs or no declining trend may be discovered, which may alter the conclusions. This is also the reason why we tested different criteria of ODEs in this study and compared the conclusions achieved. We have expanded the discussion in the revised manuscript to address these issues such as the potential health benefits resulting from the reduced frequency of ODEs. Please see lines 337-343, 422-424 and 426-428.

Regarding the reviewer's suggestion to involve more observational data such as BrO to support our conclusions, we fully agree with the reviewer that additional observational data, particularly BrO measurements, would greatly enhance our analysis and provide more robust evidence to support our conclusions. Unfortunately, we currently do not have access to such data. At present, we are actively seeking opportunities to collaborate with other research groups and institutions to obtain a broader range of observational data in the future. In the meantime, we will clearly acknowledge the limitations of our current dataset in the manuscript and emphasize the importance of obtaining additional data to further support our findings and assumptions. Please see lines 451-454 in the revised manuscript.

**Q2.2 The trend discussed in Section 3.1 and Section 3.3 are opposite, the discussion**

**throughout the paper should be more rigorous and unified.**

**A2.2** Thank you very much for the comments. We carefully checked the discussions in Section 3.1 and Section 3.3, and found the statements in Section 3.1 inaccurate which may have led to some confusion of the reviewer.

In Section 3.1 we described the interannual variability of ODE hours generally, because the focus of this section is the comparison of results between different criteria. In this section, we mentioned that there seems to have a *weak* upward trend in ODE hours between 2000 and 2012 if the year 2000 is excluded, suggesting a *slight* increase in ODE hours from 2000 to 2012. However, after the year 2012 (from 2012 to 2022), a *significant* decrease in ODE hours is indicated. In Section 3.1 of the original manuscript, we did not mention the general trend of ODE hours between the year 2000 and 2022, which may confuse the reviewer. Actually, a general decline in ODE occurrence frequency over this 23-year period can be observed from the results shown in Fig. 3(a) in the manuscript.

In Section 3.3, we provided a more quantitative analysis of the temporal behavior of ODEs, focusing on the monthly and yearly variability of ODE hours at BRW. We found April as the predominant month for ODEs and revealed a significant decrease in ODE hours over these 23 years, especially when using more stringent or relative thresholds. This section complements the broader interannual trends discussed in Section 3.1 by providing a finer temporal resolution and a more detailed examination for specific months and criteria.

In order to clarify the confusion of the reviewer, we have added more explanations in Section 3.1. Please see lines 183-184. We hope the confusion of the reviewer can be clarified after making these modifications.

**Q2.3 In section 3.2, the authors should add all the other tests' results in SI, other than the TM1, TM4, VM, and IF presented in main text.**

**A2.3** Thanks for the comment. According to the suggestion, we added the results of all other tests (TM1-5 ppbv, TM1-4 ppbv, TM2, TM3, and TM5) to the Supplementary Information. Please see Figs. S2 and S3 in the revised Supplementary Information. We hope it can provide a more comprehensive view of various criteria we used to identify ODE hours and their respective outcomes.

**Q2.4 A map of station location is necessary, especially when the author discuss the relationship between meteorological condition (such as wind directory) and ODEs event.**

**A2.4** Thank you very much for the suggestion. We added a detailed map of the station location into the revised version of the manuscript (see Fig. 1 in the manuscript). This map clearly shows the location of the BRW station at Utqiaġvik in the Arctic and includes key geographical features such as the Arctic Circle and major continents.

**Q2.5 The font in all figures should be of same size, there are some figures having too-small font, such as Figure 3, 5 and 6 etc... I suggest authors to replot all figures, all of which are not clear enough, and the styles are more like a report, not paper.**

**A2.5** Thank you for the suggestions on the font size and the clarity of figures in our manuscript. We have thoroughly revised all the figures to ensure consistency and clarity. We have also enlarged the fonts in Figures 3, 5, and 6 to make them more legible. Please see all the new figures in the revised manuscript. If the reviewer have any further recommendations or additional feedback on the improvements of the figures, we will continue to improve them until they are clear enough. Thanks again for the comment.

**Specific comments:**

**Q2.6 Line 20, the reason why the ODE only happen in spring should be explained in the Introduction part.**

**A2.6** Thanks. We have included an explanation for why ODEs predominantly occur in the spring in the introduction section of the revised manuscript. The revised text now explains that a unique combination of meteorological and chemical conditions in spring, including the presence of sunlight, strong temperature inversion, and the availability of halogen ions from ice/snow surfaces, creates the ideal environment for ODEs to occur. In contrast, these conditions are not simultaneously met in other seasons, which explains the seasonal occurrence of ODEs.

We added the following text into the Introduction section (lines 54-61 in the revised manuscript):

"The occurrence of ODEs is predominantly confined to the spring season due to the unique combination of meteorological and chemical conditions (Lehrer et al, 2004). First, the sunlight during spring is essential for photochemical reactions to take place, which is crucial for converting inert halogen ions into reactive halogens. Second, the strong temperature inversion that forms in the spring effectively isolates the boundary layer air from the free troposphere, preventing the downward mixing of ozone-rich air from aloft, allowing the reactive halogens to efficiently deplete the local ozone. Additionally, the snowpack above the sea ice in springtime acts as a significant source of halogen ions, as the brine layer on the sea ice is enriched with halogens that can be released into the atmosphere through photochemical processes. These factors collectively create the ideal conditions for ODEs to occur in the spring, while such conditions are not simultaneously met in other seasons."

**Q2.7 I suggest the author to move the Figure 1 to SI, and present the original O3 concentration trend since 2010 instead.**

**A2.7** Thanks for the suggestion. We have moved the original Fig. 1 to the Supplementary Information as Fig. S1. In place of the original Fig. 1, we present the figure showing the original $O_3$ concentrations in different months from the year 2000 to 2022 (see Fig. A1 in this rebuttal). This figure provides a direct view of the ozone changes over the past decade, which we believe will be more informative for the readers.

[Figure]

Figure A1 Monthly averaged ozone mixing ratios at the BRW Station for spring months (March, April, and May) from the year 2000 to 2022

**Q2.8 The Lines in Figure 2 is not easy to distinguish, I suggest authors to classified them into two panels, such as traditional results in one, and the rest methods in another one. The color of lines should reflect the method clusters. The current color setting is too hard to follow.**

**A2.8** Thanks a lot for the suggestion. We modified Figure 2 according to the reviewer's suggestion (see Fig. A2 in this rebuttal). In the present version of the manuscript, the figure is divided into two subplots. The upper subplot presents the results of the TM series methods, while the lower subplot shows the results of the VM and IF methods. In addition, we have re-ordered the curves in each subplot according to the mean values of the criteria. Moreover, we have adjusted the colors of the lines to better reflect the method clusters, ensuring that the current color setting is more intuitive and easier to follow. We hope these changes will address your concerns and enhance the clarity of the figure.

[Figure]

Figure A2 Number of ODE hours identified by each criterion from 2000 to 2022

**Q2.9 Line 200, the IF curve and VM curve are not alike at all in Figure 2. How do the authors have this conclusion: "it is seen that after the year 2014, the IF curve behaves**

**similarly to those of the TM methods, while before 2014, the IF curve's trend is more like the VM method's trend"?**

**A2.9** Thank you for the comment. Upon re-examining Figure 2, we agree that our initial conclusion regarding the similarity between the IF curve and the VM curve was not that accurate. We appreciate the reviewer for pointing it out.

We have modified the text to more accurately reflect the observed trends in the data. Specifically, we changed the number "2014" to "2008", as the IF curve's trend before 2008 is more similar to the VM method's trend. The updated text now clearly states these observations without any misleading statements. Thanks again for pointing this issue out.

"With respect to the Isolation Forest method (see the IF curve in Fig. 3b), generally, the number of ODE hours screened by this method is comparable to those picked out using TM methods. Interestingly, it is seen that after the year 2014, the IF curve behaves similarly to those of the TM methods, while before 2008, the IF curve's trend is more like the VM method's trend, although the values are significantly higher, indicating a possible consideration of the standard deviation in the IF method. Because of the black box characteristics of machine learning models (Hassija et al., 2024), it is difficult for us to further explore the reasons and principles behind the screening results of this method. Further interpretability of this machine learning method is also one of the areas we aim to investigate in the future."

**Q2.10 Line 235. Add results in 2013 and 2022 in SI.**

**A2.10** We have added the results for 2013 and 2022 in the Supplementary Information (SI) in the revised version of the manuscript. Please see Figs. S4(a) and (b) in the revised Supplementary Information. Thanks.

**Q2.11 Line 265, the conclusion of "the machine learning approach exhibits a limitation in accurately identifying ODE hours in years characterized by a high frequency of ODE occurrences" is too arbitrary, since the authors only tried one ML method.**

**A2.11** Thanks for the suggestion. The reviewer is correct that our conclusion was overly broad, as we only tested one machine learning method, the Isolation Forest (IF). We have changed the text as follows to be more precise.

"Thus, the IF method exhibits a limitation in accurately identifying ODE hours in years characterized by a high frequency of ODE occurrences."

Please also see line 295 in the revised manuscript.

**Q2.12 How is regression done in Figure 5?**

**A2.12** The reviewer doubted about the linear regression analysis shown in Figure 5. The specific steps and details of the regression analysis are as follows:

To assess the trend of ozone concentration over time, we employed linear regression analysis. Linear regression is a classical statistical method used to establish a linear relationship between a

dependent variable (such as ODE hours) and one or more independent variables (such as the year). Through the method of least squares, the linear regression model can find the best-fit line to quantify the trend of the dependent variable with respect to the independent variable (Draper & Smith, 1998; Weisberg, 2005; Montgomery et al., 2021).

The general form of the linear regression model is:

$$Y=aX+b,$$

where Y is the dependent variable (ODE hours in this study). X is the independent variable (year in this study). The slope a is the regression coefficient, representing the rate of change of Y with X. The intercept b represents the value of Y when X=0.

For computational convenience, we transformed the variable X (i.e., year) into its difference from the year 2000 (i.e., X-2000). Thus, the model can be written as:

$$Y=a(X-2000)+b,$$

where the slope a and the intercept b are determined by data fitting.

To evaluate the significance of the regression model, we also calculated the p-value. The p-value is usually used to judge whether the regression is significantly or not, with $p<0.05$ typically indicating that the regression is statistically significant.

$$a = \frac{\sum_{i-1}^{n}(X_i - \bar{X})(Y_i - \bar{Y})}{\sum_{i-1}^{n}(X_i - \bar{X})^2}$$

$$b = \bar{Y} - a\bar{X}$$

$$SE(a) = \frac{\sum_{i-1}^{n}\frac{(Y_i - \bar{Y})^2}{n-2}}{\sum_{i-1}^{n}(X_i - \bar{X})^2}$$

$$t = \frac{a}{SE(a)}$$

$$\text{p-value} = 2 \times P(T>|t|)$$

Where $\bar{X}$ and $\bar{Y}$ are the means of X and Y, respectively. Where a is the regression coefficient and SE(a) is its standard error. Where T is the t-distribution random variable with $df=n-2$ degrees of freedom, and P represents the probability. The results shown in the present manuscript indicate that the trend of ozone concentration over time is statistically significant ($p<0.05$) or highly significant ($p<0.01$).

We added a sub-section describing the linear regression used in the revised manuscript. Please see the Section 2.3 (lines 161-174) in the revised manuscript .

**References**

Burd, J. A., Peterson, P. K., Nghiem, S. V., Perovich, D. K., and Simpson, W. R.: Snowmelt onset hinders bromine monoxide heterogeneous recycling in the Arctic, Journal of Geophysical Research: Atmospheres, 122, 8297–8309,https://doi.org/https://doi.org/10.1002/2017JD026906, 2017.

Draper, N. R. and Smith, H.: Applied regression analysis, John Wiley & Sons, 326 pp., 1998.

Lehrer, E., Hönninger, G., and Platt, U.: A one dimensional model study of the mechanism of halogen liberation and vertical transport in the polar troposphere, Atmospheric Chemistry and Physics, 4, 2427–2440, https://doi.org/10.5194/acp-4-2427-2004, 2004.

Montgomery, D. C., Peck, E. A., and Vining, G. G.: Introduction to linear regression analysis, John Wiley & Sons, 2021.

Weisberg, S.: Applied linear regression, John Wiley & Sons, 528 pp., 2005.